# Determinants of first-line antiretroviral treatment failure among adult patients on treatment in Mettu Karl Specialized Hospital, South West Ethiopia; a case control study

Sabit Zenu[1]ᴼ*, Tariku Tesema[2]ᴼ, Mohammed Reshad[1]ᴼ, Endegena Abebe[3]ᴼ

1 Department of Public Health, College of Health Sciences, Mettu University, Mettu, Ethiopia, 2 Mettu College of Health Sciences, Mettu, Ethiopia, 3 Department of Biomedical Sciences, College of Health Sciences, Mettu University, Mettu, Ethiopia

ᴼ These authors contributed equally to this work.
* sabitzeinu91@gmail.com

**Data Availability Statement:** All relevant data are within the manuscript and its Supporting information files.

## Abstract

### Background

Antiretroviral therapy has dramatically reduced Human Immunodeficiency Virus related morbidity and mortality. It has also transformed HIV infection into a manageable chronic condition. However, first-line antiretroviral treatment failure continues to grow especially in resource limited settings. Despite this, determinants of first-line antiretroviral treatment failure are not well studied in Ethiopia.

### Objective

To identify determinants of first-line antiretroviral treatment failure among adult patients on antiretroviral therapy in Mettu Karl Specialized Hospital, South West Ethiopia, in 2020.

### Methods

A hospital based case-control study was conducted from October to November 2020. Simple random sampling technique was used to select participants. Interviewer administered questionnaire and record review were used for data collection. Data were entered into epi data version 3.1 and exported to SPSS version 20 for analysis. Bivariable and multivariable logistic regression analysis were used. At the end, variables with P-value < 0.05 at 95% confidence intervals for adjusted odds ratio were considered statistically significant determinants of first line treatment failure.

### Result

A total of 113 cases and 339 controls were included in the study with response rate of 98.6%. Sixty-four (56.6%) of cases and 183 (54.0%) of controls were females. Baseline WHO clinical stage III and IV (AOR = 1.909, 95% CI: (1.103, 3.305), baseline body mass index<18.5kg/m$^2$(AOR = 2.208,95% CI:(1.257,3.877),baseline CD4 cell count <100cells/

**Funding:** The funders had no role in study design, data collection and analysis, decision to publish, or preparation of the manuscript.

**Competing interests:** The authors have declared that no competing interests exist.

$mm^3$ (AOR = 3.016, 95% CI: (1.734, 5.246), having history of TB co-infection (AOR = 1.855, 95% CI: (1.027, 3.353), having history of lost to follow up (AOR = 3.235, 95% CI: (1.096, 9.551), poor adherence to medication (AOR = 7.597, 95% CI: (4.059, 14.219) and initiation of treatment after two years of diagnosis with HIV (AOR = 4.979, 95% CI: (2.039, 12.158) were determinants of first-line antiretroviral treatment failure.

## Conclusion

In this study several variables were found to be determinants of first-line antiretroviral treatment failure. Concerned bodies should give more attention to early diagnosis of HIV, early enrollment in chronic HIV care and early initiation of ART before patients develop advanced WHO clinical stages. In addition, focus has to be given for patients with low CD4 count. Regular screening for TB, counseling on optimal adherence to medication and enhancing nutritional status of patients with low body mass index are also crucial to prevent first-line antiretroviral treatment failure.

## Introduction

Human Immunodeficiency Virus (HIV) and its clinical outcome, Acquired Immune Deficiency Syndrome(AIDS), has become one of the world's major public health problems and development challenges [1]. According to the Joint United Nations Program on HIV/AIDS (UNAIDS) report, an estimated 37.7 million people were living with HIV/AIDS in 2020. In the same year, 1.5 million people became newly infected with HIV and over 680,000 people died of the disease. From the total people living with HIV, 27.5 million were accessing Anti-Retroviral Therapy (ART). From the total of over 79.3 million infections since the start of the pandemic, 36.3 million people have died so far [2].

ART has dramatically reduced HIV related morbidity and mortality and has transformed HIV infection into a manageable chronic condition. The treatment is also highly effective at reducing sexual transmission of HIV in patients who have adequately suppressed viral loads [3]. The ART treatment goals are suppression of HIV replication, restoration and preservation of immune function, reduction in HIV related morbidity and mortality that improves the quality of life of People Living With HIV/AIDS (PLWHA) [4]. On the other hand, when first-line antiretroviral treatment failure develops, all benefits of ART are affected [5].

Virological suppression, clinical recovery and immunological improvement are expected from PLWHA after initiation of ART [3]. First-line ART failure occurs when a combination of the antiretroviral regimen fails to control HIV infection. This could be virologic, immunologic and/or clinical failure [6]. Virologic failure occurs when plasma viral load remains above 1000 copies/ml based on two consecutive viral load measurements in three-month interval, with adherence support following the first-viral load test. Immunological failure occurs when CD4 count is at or below 250 cells/$mm^3$ following clinical failure or persistent CD4 levels below 100cells/$mm^3$. Clinical failure occurs when new or recurrent clinical event indicating severe immunodeficiency (World Health Organization(WHO) clinical stage IV condition) occurs after six months of effective treatment [4].

Frequent assessment of treatment response is important while the patient is on ART. Monitoring the response to ART and diagnosis of treatment failure for patients on antiretroviral therapy is important to achieve treatment goals. First-line antiretroviral treatment failure can

be assessed virologically, immunologically and or clinically. The WHO recommends viral load monitoring as a backbone for detection of treatment failure [5].

The WHO recommended viral load testing as a preferred monitoring approach to diagnose and confirm antiretroviral treatment failure in 2013. However, in low- and middle-income countries, where the majority of individuals living with HIV live, viral load testing is scarce (7). When compared to clinical and immunological surveillance, viral load testing provides an earlier and more accurate indication of therapy failure (4).

Globally, about 10–20% of adult patients on first-line antiretroviral treatment are reported to have developed treatment failure with higher figures (15–25%) being reported in Sub-Saharan Africa [7]. In Sub-Saharan Africa, many patients who experience treatment failure do not switch to potent second-line regimens due to resource limitation, yet those who remain on failing first-line regimen experience disproportionately higher morbidity and mortality compared to those who switch [8]. In Ethiopia, prevalence of first-line antiretroviral treatment failure was 15.3% by using the three WHO treatment failure criteria (virological, immunological and clinical) [9].

Despite the scaling up of antiretroviral treatment in resource limited settings, development of first-line antiretroviral treatment failure remained a big challenge [10]. Treatment failure among population taking ART in Ethiopia is a public health concern because patients experiencing treatment failure will have an increased risk of morbidity, mortality and increased transmission as well as accumulation of drug resistant mutations [11].

According to a study conducted in the United States(US) in 2014, the cost of treating a patient with a second-line ART drug increases by 24% as compared to the first-line treatment [12]. Currently, in Ethiopia where medication is fully funded by the government, treatment failure and frequent substitution of medications are becoming major challenges in control of the disease [13].

Different studies identified that age <35 years, being male, higher educational level, urban residence, unemployment, advanced clinical stage III/IV, having history of TB co-infection, baseline CD4< 100cell/mm3, baseline BMI <18.5 kg/mm2, poor adherence, lost to follow up, baseline ART regimen, high frequency of alcohol use and smoking as determinant factors of first-line antiretroviral treatment failure [14–18].

Investigating and managing determinant factors of first-line antiretroviral treatment failure is very important to achieve treatment targets, decrease morbidity and mortality, decrease HIV transmission and sustain the quality of life of PLWHA.

The need of undertaking the study is that there is limited evidence on determinants of first-line antiretroviral treatment failure in Ethiopia and no known research has been done in the study area to identify determinant factors of first-line ART failure among adult patients on treatment. Even though viral load test is the gold standard technique for early detection of first-line antiretroviral treatment failure, most of the previous studies in Ethiopia, did not consider virological failure because of the absence of the test service in primary care settings. Many studies considered only immunological and clinical failure as criteria of first-line antiretroviral treatment failure. But in this study virological failure, which is one of the decision criteria of first-line antiretroviral treatment failure was considered in addition to immunological and clinical failure. Therefore, this study aimed to identify determinants of first-line antiretroviral treatment failure among adult patients on antiretroviral therapy in Mettu Karl Specialized Hospital, South West Ethiopia.

## Materials and methods

### Study design, area and period

A hospital based unmatched case-control study was conducted. The study was conducted at Mettu Karl Specialized Hospital which is the only specialized hospital in the area serving a

population of more than 1.4 million with different services including HIV prevention, care and treatment. The hospital is located in Mettu town, on a distance of 600 Kilometers from Ethiopian capital Addis Ababa. It provides specialized health care for peoples in three neighboring regions of Ethiopia; Oromia, Gambella and South Nations Nationalities and Peoples Regions. The Gambella Region is one of Ethiopia's nine regions with the highest HIV prevalence and incidence. In this hospital, ART service started in 2005. Currently total of 1600 people are receiving ART at the facility. The hospital has laboratory services to determine CD4 count and viral load to monitor ART patients. The study was conducted from October 25 to November 24/2020.

## Source and study populations

**Source populations.** The source populations for cases in the study were all adult PLWHA documented to have first-line antiretroviral treatment failure and enrolled to the second-line antiretroviral treatment at the hospital. The source populations for controls were all adult PLWHA who did not develop first-line antiretroviral treatment failure and on first-line antiretroviral treatment the hospital.

**Study populations.** Study populations for cases in the study were adult HIV Patients on ART, aged $\geq$ 18 years, documented to have first-line antiretroviral treatment failure and eligible for the study during study period. Study population for controls were adult HIV patients on ART, aged $\geq$18 years, who did not fail first-line antiretroviral treatment and who were on first-line ART for six or more months.

## Sample size determination and sampling technique

Sample size was calculated using Epi Info version 7 for unmatched case control study design by considering the significant determinant factor of first-line antiretroviral treatment failure such as baseline BMI< 18.5kg/m$^2$, having history of lost to follow up, baseline CD4 cell<50 cell/mm$^3$, poor adherence, smoking, history of TB co-infection(Table 1).

From the above table, poor adherence was found to result in the largest sample size; it was used to determine sample size as independent variable. By using 95% Confidence Interval(CI), 80% power, a case to control ratio of 1:3 and using a two population proportion formula, the calculated sample size was 416 (104 cases and 312 controls). Then by adding 10% for non-response, the final sample size was 458 (115 cases and 343 controls).

Simple random sampling technique was used to recruit study participants for cases and controls. Adult patients who developed first-line antiretroviral treatment failure and those who did not develop first-line antiretroviral treatment failure were identified and their Medical Record Numbers (MRN) was listed as cases and controls. Accordingly, register of 1353 patients who did not develop first-line antiretroviral treatment failure and on first-line was

**Table 1. Sample size calculation for the study on determinants of first-line antiretroviral treatment failure among adult patients on antiretroviral therapy in Mettu Karl Specialized Hospital, South West Ethiopia, 2020.**

| Variables | The ratio of cases to controls | % of Controls exposed | AOR | % of Cases exposed | Sample size |
|---|---|---|---|---|---|
| Baseline BMI< 18.5 kg/m$^2$ [19] | 1:3 | 44.5 | 2.75 | 68.8 | 194(49, 145) |
| Having history of lost to follow up [20] | 1:3 | 6.01 | 3.66 | 19.0 | 282(71, 211) |
| Baseline CD4 < 50 cells/ mm$^3$ [21] | 1:3 | 10.1 | 3.96 | 30.5 | 167(42, 125) |
| Poor adherence [17] | 1:3 | 6.08 | 2.986 | 16.2 | **416(104, 312)** |
| Smoking [22] | 1:3 | 18 | 5.9 | 56.4 | 72 (18, 54) |
| History of TB co-infection [22] | 1:3 | 26.7 | 3.9 | 58.7 | 110(28, 82) |

developed. Then to select study subjects for controls simple random sampling technique was used by computer generated random numbers based on MRN of patients.

For cases, register of 194 adult patients who developed first-line antiretroviral treatment failure and enrolled to the second-line antiretroviral therapy at the hospital was prepared. Cases were then selected by simple random sampling technique using computer generated random numbers based on MRN of patients.

### Data collection tools and procedures

Data was collected from patients and medical records using structured interviewer administered questionnaire and structured checklists respectively. The questionnaire was developed from WHO ART guideline, ART follow up guideline of Federal Ministry of Health of Ethiopia and Ethiopian Demographic and Health Survey(EDHS) 2016 [4,5,23]. In addition, other tools were adopted from previously conducted studies [19,21,24,25]. The questionnaire comprised four parts; socio-demographic factors, clinical factors, antiretroviral treatment related factors and behavioral factors. Questions related with behavioral factors and socio-demographic factors were included in structured interviewer administered questionnaires and collected from patients. Questions related to clinical and antiretroviral treatment related factors were included in the checklist and collected from record review. In the hospital, CD4 T- cell count measurement was done by using the Pima™ CD4 Analyzer (Abbott Molecular Inc., Chicago, Illinois, United States, formerly and Alere). The Viral load measurement was conducted by using the Roche (COBAS® AmpliPrep/COBAS® TaqMan® HIV-1 Test, Roche Diagnostics, Indianapolis, Indiana, United States). All laboratory procedures were conducted according to the Ethiopian national protocol for laboratory tests for ART patients [26].

### Operational definitions

**Adults**: In this study, participants aged 18–64 years are considered as adults [23].

**Poor adherence**: Less than 85% adherence, which is defined as missing greater than or equal to five doses out of 30 doses or greater than 10 doses from 60 doses.

**Fair adherence**: 85%-94% adherence, which is defined as missing two to four doses out of 30 doses or four to nine doses from 60 doses.

**Good adherence**: Greater than or equal to 95% adherence i.e. missing less than or equal to one out of 30 doses or missing less than or equal to two from the 60 prescribed doses [5,27].

### Data analysis procedures

The collected data was coded and entered into epi data version 3.1 and exported to statistical package for social sciences (SPSS) version 20 for analysis. Checking and cleaning of data was done before analysis. Frequencies and proportions were used to describe the data. Cross tabulation was used to summarize descriptive statistics in each group. Bivariable and multivariable logistic regression were used to identify determinants of first-line antiretroviral treatment failure. Variables with P-value <0.25 at bivariable analysis were taken to multivariable logistic regression analysis. The multivariable model was fitted to identify the independent determinants of first-line antiretroviral treatment failure using backward stepwise removal method. The model fitness was checked by Hosmer and Lemeshow goodness of fit test by considering p-value >0.05. Finally, variables that have significant association with first-line antiretroviral treatment failure were identified and reported based on the adjusted odds ratio (AOR) with corresponding 95% CI at P-value <0.05.

**Ethics statement.** This research is undertaken in full compliance to the ethics requirements of the Declaration of Helsinki. The proposal of the research work was reviewed and

approved by Ethics Review Committee of College of Health Sciences, Mettu University (Ref. No RCS/050/2020). Written informed consent was taken from individual participants after detailed explanation of the research procedures. Confidentiality of the research participant's information was kept and no identifying information was included in the data collection tool.

## Result

### Socio-demographic characteristics of participants

A total of 452(113 cases and 339 controls) HIV patients on ART were involved in the study with response rate of 98.6%. Sixty-four (56.6%) of cases and 183 (54.0%) of controls were females. Regarding marital status, 42 (37.2%) of cases and 193 (56.9%) of controls were married; while 30 (26.5%) of cases and 33 (9.7%) of controls were single. Forty-one (36.3%) of cases and 92 (27.1%) of controls were living in rural areas (Table 2).

### Bivariable logistic regression analysis of first-line antiretroviral treatment failure

Bivariable logistic regression analysis was carried out to assess the association of variables with first-line treatment failure. Among these variables, smoking, khat chewing, baseline WHO

**Table 2. Socio-demographic characteristics of HIV patients on ART at Mettu Karl Referral Hospital, South West Ethiopia, 2020.**

| Variable | Category | Cases (%) (n = 113) | Control (%) (n = 339) | Total (%) |
|---|---|---|---|---|
| Age at initiation of ART | < 35 years | 49 (43.4%) | 119 (35.1%) | 168 (37.2%) |
| | > = 35 years | 64(56.6%) | 220 (64.9%) | 284 (62.8%) |
| Sex | Male | 49 (43.4%) | 156 (46.0%) | 205 (45.4%) |
| | Female | 64 (56.6%) | 183 (54.0%) | 247 (54.6%) |
| Marital status | Married | 42 (37.2%) | 193 (56.9%) | 235 (52%) |
| | Single | 30 (26.5%) | 33 (9.7%) | 63 (13.9%) |
| | Divorced | 25 (22.1%) | 63 (18.6%) | 88 (19.5%) |
| | Widowed | 16 (14.2%) | 50 (14.7%) | 66 (14.6%) |
| Place of residence | Urban | 72 (63.7%) | 247 (72.9%) | 319 (70.6%) |
| | Rural | 41 (36.3%) | 92 (27.1%) | 133 (29.4%) |
| Occupational status | Government employee | 13 (11.5%) | 37 (10.9) | 50 (11.1%) |
| | Farmer | 20 (17.7%) | 45 (13.3%) | 65 (14.4%) |
| | Daily laborer | 21 (18.6%) | 57 (16.8%) | 78 (17.3%) |
| | Merchant | 21 (18.6%) | 79 (23.3%) | 100 (22.1%) |
| | House wife | 22 (19.5%) | 95 (28%) | 117 (25.9%) |
| | Others* | 16 (14.2%) | 26 (7.7%) | 26 (5.8%) |
| Educational status | No formal education | 21 (18.6%) | 60 (17.7%) | 81 (17.9%) |
| | Primary school (1–8) | 41 (36.3%) | 169 (49.9%) | 210 (46.5%) |
| | Secondary school (9–12) | 33 (29.2%) | 67 (19.8%) | 100 (22.1%) |
| | College and above | 18 (15.9%) | 43 (12.7%) | 61 (13.5%) |
| Criteria used to diagnose treatment failure | Virological | 24 (21.2%) | | |
| | Clinical/immunological | 39 (34.5%) | | |
| | Clinical/virological | 8 (7.1%) | | |
| | Immunological/virological | 7 (6.2%) | | |
| | All criteria | 35 (31.0%) | | |

*Students, driver, self and private business, Non-Governmental Organization workers.

**Table 3. Bivariate logistic regression analysis of first-line antiretroviral treatment failure of HIV patients on ART at Mettu Karl Specialized Hospital, South West Ethiopia, 2020.**

| Variable | Category | Case (N = 113) | Control (N = 339) | COR (95% CI) | P-value |
|---|---|---|---|---|---|
| Smoking status | Yes | 28 (24.8%) | 57 (16.8%) | 1.630 (0.975, 2.723) | .062 |
| | No | 85 (75.2%) | 282 (83.2%) | 1 | |
| Khat chewing | Yes | 30 (26.5%) | 67 (19.8%) | 1.462 (0.890, 2.400) | .133 |
| | No | 83 (73.5%) | 271 (80.2%) | 1 | |
| Baseline WHO stage | Stage I and II | 53 (46.9%) | 234 (69.0%) | 1 | |
| | Stage III and IV | 60 (53.1%) | 105 (31.0%) | 2.523 (1.632, 3.899) | < .01 |
| Baseline BMI | <18.5kg/m$^2$ | 63 (55.8%) | 89 (26.3%) | 3.539 (2.272, 5.513) | < .01 |
| | > = 18.5kg/m$^2$ | 250 (73.7%) | 50 (44.2%) | 1 | |
| Baseline CD4 cell count | <100cells/mm$^3$ | 59(52.2%) | 69 (20.4%) | 4.275 (2.715, 6.732) | < .01 |
| | > = 100cells/mm$^3$ | 54 (47.8%) | 270 (79.6%) | 1 | |
| TB co-infection | Yes | 46 (40.7%) | 65 (19.2%) | 2.894 (1.822, 4.596) | < .01 |
| | No | 67 (59.3%) | 274 (80.8%) | 1 | |
| History of lost to follow up | Yes | 15 (13.3%) | 8 (2.4%) | 6.333 (2.608, 15.378) | < .01 |
| | No | 98 (86.7%) | 331 (97.6%) | 1 | |
| Adherence status | Good | 37 (32.7%) | 240 (70.8%) | 1 | |
| | Fair | 18 (15.9%) | 65 (19.2%) | 1.796 (0.960, 3.360) | .067 |
| | Poor | 58 (51.3%) | 34 (10.0%) | 11.065 (6.404, 19.118) | < .01 |
| Disclosure status | Yes | 105(92.9%) | 334 (98.5%) | 1 | |
| | No | 8 (7.1%) | 5 (1.5%) | 5.090 (1.630, 15.893) | .005 |
| Time lag to initiate ART after diagnosis with HIV | Within the same month | 30 (26.5%) | 158 (46.6%) | 1 | |
| | One to twenty four months | 65 (57.5%) | 157 (46.3%) | 2.180 (1.342, 3.544) | .002 |
| | After twenty four months | 18 (15.9%) | 24 (7.1%) | 3.950 (1.913, 8.157) | < .01 |

COR: Crude odds ratio; CI: Confidence interval; 1: Reference category.

clinical stage, baseline BMI, baseline CD4 count, history of TB co-infection, history of lost to follow up, adherence status to antiretroviral drugs, disclosure status, time lag to initiate ART after diagnosis with HIV were candidates for multivariable logistic regression analysis at P-value <0.25 in bivariable logistic regression model (Table 3).

## Determinants of first-line antiretroviral treatment failure

In multivariable logistic regression analysis, baseline WHO clinical stage III and IV, baseline body mass index < 18.5kg/m$^2$, baseline CD4 count <100 cells/mm$^3$, having history of TB co-infection, having history of lost to follow up, poor adherence to antiretroviral drugs, initiation of ART after two years of diagnosis with HIV were significantly associated with first-line antiretroviral treatment failure.

The finding of this study showed that, HIV positive patients on ART with stages III and IV baseline WHO clinical stage were almost two times more likely to fail first-line antiretroviral treatment when compared with stage I and II baseline WHO clinical stage [AOR = 1.909, 95% CI: 1.103, 3.305].

Patients on ART with baseline Body Mass Index of <18.5kg/m$^2$ were two times more likely to develop first-line antiretroviral treatment failure than patients with baseline BMI of > = 18.5kg/m$^2$ [AOR = 2.208, 95% CI: 1.257, 3.877].

Patients on ART withCD4 count of <100cells/mm$^3$ at the time of initiation of ART had three times more likelihood to fail first-line antiretroviral treatment when compared with

**Table 4. Multivariable logistic regression analysis on determinants of first-line antiretroviral treatment failure among HIV Patients on ART at Mettu Karl Specialized Hospital, South West Ethiopia, 2020.**

| Variable | Category | Case (N = 113) | Control (N = 339) | COR (95% CI) | AOR (95% CI) |
|---|---|---|---|---|---|
| Baseline WHO clinical stage | Stage I and II | 53 (46.9%) | 234 (69.0%) | 1 | 1 |
| | Stage III and IV | 60 (53.1%) | 105 (31.0%) | 2.523 (1.632, 3.899) | **1.909 (1.103, 3.305)**[*] |
| Baseline BMI | <18.5kg/m$^2$ | 63 (55.8%) | 89 (26.3%) | 3.539 (2.272, 5.513) | **2.208 (1.257, 3.877)**[*] |
| | > = 18.5kg/m$^2$ | 250 (73.7%) | 50 (44.2%) | 1 | 1 |
| Baseline CD4 | <100 Cells/mm3 | 59(52.2%) | 69 (20.4%) | 4.275 (2.715, 6.732) | **3.016 (1.734, 5.246)**[*] |
| | > = 100Cells/mm$^3$ | 54 (47.8%) | 270 (79.6%) | 1 | 1 |
| TB co-infection | Yes | 46 (40.7%) | 65 (19.2%) | 2.894 (1.822, 4.596) | **1.855 (1.027, 3.353)**[*] |
| | No | 67 (59.3%) | 274 (80.8%) | 1 | 1 |
| History of lost to follow up | Yes | 15 (13.3%) | 8 (2.4%) | 6.333(2.608, 15.378) | **3.235(1.096, 9.551)**[*] |
| | No | 98 (86.7%) | 331 (97.6%) | 1 | 1 |
| Adherence status | Good | 37 (32.7%) | 240 (70.8%) | 1 | 1 |
| | Fair | 18 (15.9%) | 65 (19.2%) | 1.796 (0.960, 3.360) | 1.322 (0.643, 2.716) |
| | Poor | 58 (51.3%) | 34 (10.0%) | 11.065(6.404, 19.118) | **7.597 (4.059, 14.219)**[*] |
| Disclosure status | Yes | 105 (92.9%) | 334 (98.5%) | 1 | 1 |
| | No | 8 (7.1%) | 5 (1.5%) | 5.090 (1.630, 15.893) | 1.663 (0.460, 6.013) |
| Time lag to initiate ART after diagnosis | Within the same month | 30 (26.5%) | 158 (46.6%) | 1 | 1 |
| | One to twenty four months | 65 (57.5%) | 157 (46.3%) | 2.180 (1.342, 3.544) | 1.702 (0.928, 3.121) |
| | After twenty four months | 18 (15.9%) | 24 (7.1%) | 3.950 (1.913, 8.157) | **4.979 (2.039, 12.158)**[*] |

[*]Statistically significant at p-value<0.05;

COR: Crude odds ratio; AOR: Adjusted odds ratio; CI: Confidence interval; 1: Reference category.

those who had CD4 count of > = 100cells/mm$^3$ at the time of initiation of ART [AOR = 3.016, 95% CI: 1.734, 5.246].

Patients with history of TB co-infection were almost twice more likely to develop first-line treatment failure than those patients without TB co-infection [AOR = 1.855, 95% CI: 1.027, 3.353].

Patients with history of lost to follow up had three times more likely to fail first-line antiretroviral treatment when compared with patients did not have lost to follow up history [AOR = 3.235, 95% CI: 1.096, 9.551].

HIV positive patients on ART with poor adherence to antiretroviral drugs were more than seven times more likely to fail first-line antiretroviral treatment when compared to patients with good adherence to antiretroviral drugs [AOR = 7.597, 95% CI: 4.059, 14.219].

Patients who initiated ART after two years of being HIV positive had around five times more probability to fail first-line antiretroviral treatment when compared to patients initiated ART within the same month of being HIV positive [AOR = 4.979, 95% CI: 2.039, 12.158] (Table 4).

## Discussion

The identification and management of first-line ART failure is a key challenge for HIV programs in resource limited settings. Staying on a failing first-line antiretroviral therapy is associated with an increased risk of mortality. In addition to this, development of drug resistance limits the ability to construct new, potent and tolerable regimens in the future. This study was aimed to identify determinants of first-line antiretroviral treatment failure.

In this study first-line antiretroviral treatment failure was found to be significantly associated with stage III and IV baseline WHO clinical stage of HIV, low baseline Body Mass Index ($<18.5$ kg/m$^2$), baseline CD4 count $<100$cells/mm$^3$, having history of TB co-infection, having history of lost to follow up, poor adherence to antiretroviral drugs and initiation of ART after two years of diagnosis with HIV positive.

The current study has identified that patients with advanced WHO clinical stage III and IV at the time of initiation of ART were almost two times more likely to fail first-line antiretroviral treatment when compared with stage I and II baseline WHO clinical stage. This finding has similarity with the studies conducted in Sanglah Hospital, Bali Indonesia [15], Senegal [28], and in Harar public hospitals, Eastern Ethiopia [24]. This might be due the fact that advanced WHO clinical stage of HIV disease is associated with high viral load and low CD4 cell count that compromise immunity and may negatively affect response to first-line antiretroviral treatment [29].

In this study, patients with low baseline Body Mass Index (BMI) of $<18.5$kg/m$^2$ were two times more likely to develop first-line antiretroviral treatment failure than patients with baseline BMI of $> = 18.5$kg/m$^2$. This finding is consistent with report from studies conducted in Woldia Hospital, North East Ethiopia, by 2017 [19], University of Gonder Specialized Hospital, Ethiopia, in 2019 [30]. This might be due to the fact that patients with low BMI have low nutritional status that leads to weakened immunity, blunted immune response and increased likelihood of first-line antiretroviral treatment failure [31].

The finding of this study indicated that patients with low baseline CD4 count of $<100$cells/mm$^3$ were three times more likely to fail first-line antiretroviral treatment than patients with baseline CD4 count of $> = 100$cells/mm$^3$. This finding is comparable with studies done in India by 2016 [32], in Northwestern Tanzania by 2019 [10], in Dire Dawa, Eastern Ethiopia by 2019 [33]. This finding might be due the reason that patient with low baseline CD4 cell count have a lesser immunity that may favor the occurrence of opportunistic infection and lead to clinical failure. In addition, low CD4 cell count is difficult to be replaced enough in HIV patients on ART and may lead to first-line antiretroviral treatment failure [29,34]

Having history of TB co-infection is found to be an independent determinant of first-line antiretroviral treatment failure. Patients who had history of HIV-TB co-infection were around two times more likely to fail first-line antiretroviral treatment when compared to patients who had no history of TB co-infection. Similarly studies conducted in India in 2016 [32], Gutu District, Zimbabwe by 2017 [14], Debre Markos Referral Hospital, North West Ethiopia in 2018 [35], showed that HIV-TB co-infection was independent determinant factors of treatment failure among adult patients on ART. The occurrence of tuberculosis during antiretroviral treatment has multiple effects including pills burden and drug-drug interaction which may lead to first-line treatment failure [36].

Having history of lost to follow up was associated with first-line treatment failure. Patients with history of lost to follow up were three times more likely to develop first-line treatment failure than patients without history of lost to follow up. This finding is similar with studies conducted in Harare Central Hospital, Zimbabwe by 2014 [7], in Nigerian Teaching Hospital, Nigeria by 2019 [37], in Central Ethiopia St. Luke Referral Hospital and Tulu bolo General Hospital by 2019 [20]. This might be due to the result of on-going viral replication in the absence of ART, which results in decrease of CD4 cell count and increase viral load that leads to first-line antiretroviral treatment failure [38].

Patients with poor adherence to antiretroviral drugs have more than seven times higher probability of developing first-line treatment failure than patients with good adherence to antiretroviral drugs. The finding is consistent with studies conducted in Nigerian Teaching Hospitals by 2019 [37], in Cameroon by 2016 [39], University of Gondar Teaching Hospital by 2016 [25], in Sekota, Northeast Ethiopia by 2019 [40]. This might be due to the fact that adherence

issue is the pillar for patients on ART. Patients with poor adherence to antiretroviral drugs are associated with loss of sustained viral suppression, higher risk of drug resistance that leads to first-line antiretroviral treatment failure [41].

Patients who were enrolled to ART after two years of diagnosis with HIV were nearly five times more likely to develop first-line antiretroviral treatment failure when compared with patients who were enrolled to ART within the same month of being diagnosed with HIV. This finding was in line with those studies conducted in Zimbabwe [7], in Central Ethiopia St. Luke Referral Hospital and Tulu bolo General Hospital [20]. This might be due to the possibility that patients who stay long time without initiation of ART after diagnosis face an increase in viral load and develop other opportunistic infections. This might also be due to the difficulty of viral suppression and increase CD4 cell count when the patients delay to start ART [42].

Contrary to findings from several studies [15–17,19,39], some important socio demographic variables like age, sex and educational status were not found to be significant determinants of first line ART failure in the current study. This may be due to difference in socio demographic characteristics of study participants and socio economic differences between the study settings. Most of the studies that reported the association were conducted in Asian and West African countries. Differences in the study design and sample size may have also played roles in the observed variation. Studies outside Ethiopia used cross sectional design while some studies used huge case-control ratio which may have diluted the association.

In addition, behavioral characteristics like smoking, alcohol consumption, and chat chewing were not found to be significant determinants of ART failure. This may be due to significant reduction of these detrimental behaviors as a result of continuous counseling for HIV patients as per the national guideline in Ethiopia and the study setting [5].

## Limitation

The case-control nature of this study may have resulted in difficulties in recalling of some exposures.

## Conclusion

This study showed that baseline stage III and IV WHO clinical stage of HIV, low baseline Body Mass Index ($<18.5$ kg/m$^2$), baseline CD4 count $<100$cells/mm$^3$, having history of TB co-infection, having history of lost to follow up, poor adherence to antiretroviral drugs and initiation of ART after two years of diagnosis with HIV positive were factors associated with first-line antiretroviral treatment failure. Concerned bodies should give attention to early diagnosis of HIV, early enrollment in chronic HIV care and timely initiation of ART before patients develop advanced disease condition. In addition, focus has to be given for patients with low CD4 count. Regular screening for tuberculosis, counseling on optimal adherence to medication and enhancing nutritional status of patients with low body mass index are also crucial to prevent first-line antiretroviral treatment failure.

## Supporting information

**S1 File.**
(SAV)

## Acknowledgments

We would like to thank Mettu University College of Health Sciences, Department of Public Health for giving us all necessary cooperation to undertake this study. Our gratitude also goes

to data collectors, supervisors and study participants for their commitment and cooperation. We also express our sincere thanks to our friends for their direct and indirect contribution in finalization of this research work.

## Author Contributions

**Conceptualization:** Sabit Zenu, Tariku Tesema, Mohammed Reshad, Endegena Abebe.

**Data curation:** Sabit Zenu, Tariku Tesema, Endegena Abebe.

**Formal analysis:** Sabit Zenu, Tariku Tesema, Endegena Abebe.

**Funding acquisition:** Sabit Zenu, Tariku Tesema.

**Investigation:** Sabit Zenu, Tariku Tesema.

**Methodology:** Sabit Zenu, Tariku Tesema, Mohammed Reshad, Endegena Abebe.

**Project administration:** Sabit Zenu, Tariku Tesema.

**Resources:** Sabit Zenu, Tariku Tesema.

**Software:** Sabit Zenu, Tariku Tesema, Endegena Abebe.

**Supervision:** Sabit Zenu, Tariku Tesema, Endegena Abebe.

**Validation:** Sabit Zenu, Tariku Tesema.

**Visualization:** Sabit Zenu, Tariku Tesema, Endegena Abebe.

**Writing – original draft:** Tariku Tesema, Mohammed Reshad, Endegena Abebe.

**Writing – review & editing:** Sabit Zenu, Tariku Tesema, Mohammed Reshad, Endegena Abebe.

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
