## [Decision Letter · Decision Letter 0]

26 Aug 2021

PONE-D-21-13362

Determinants of first-line antiretroviral treatment failure among adult patients on antiretroviral therapy in a Specialized Hospital, South West Ethiopia; A case control Study

PLOS ONE

Dear Dr. Siraj,

Thank you for submitting your manuscript to PLOS ONE. After careful consideration, we feel that it has merit but does not fully meet PLOS ONE’s publication criteria as it currently stands. Therefore, we invite you to submit a revised version of the manuscript that addresses the points raised during the review process.

Two experts in the field handled your manuscript, and we are appreciative of their time and contributions. Although interest was found in your study, several major concerns arose that require your attention. Notably, the novelty of this study needs to be stated in the rationale. Also, it was indicated that the scope of this work is a bit narrow to a region of Ethiopia and should be expanded to engage the broad audience of this Journal and how these findings could relate to international health outcomes. 

We look forward to receiving your revised manuscript.

Kind regards,

Frank T. Spradley

Academic Editor

PLOS ONE

Journal Requirements:

For additional information about PLOS ONE ethical requirements for human subjects research, please refer to " ext-link-type="uri" xlink:type="simple">http://journals.plos.org/plosone/s/submission-guidelines#loc-human-subjects-research."

3. We noted several instances in the manuscript where the P value equals zero; please correct and clarify.

6. Thank you for submitting the above manuscript to PLOS ONE. During our internal evaluation of the manuscript, we found significant text overlap between your submission and the following previously published works, some of which you are an author.

https://www.dovepress.com/virological-treatment-failure-among-adult-hivaids-patients-from-select-peer-reviewed-fulltext-article-IDR

https://www.researchsquare.com/article/rs-2178/v2

https://www.dovepress.com/first-line-antiretroviral-treatment-failure-and-associated-factors-in--peer-reviewed-fulltext-article-HIV

Please revise the manuscript to rephrase the duplicated text, cite your sources, and provide details as to how the current manuscript advances on previous work. Please note that further consideration is dependent on the submission of a manuscript that addresses these concerns about the overlap in text with published work.

Reviewers' comments:

Reviewer's Responses to Questions

**Comments to the Author**

1. Is the manuscript technically sound, and do the data support the conclusions?

Reviewer #1: Partly

Reviewer #2: Yes

2. Has the statistical analysis been performed appropriately and rigorously? 

Reviewer #1: Yes

Reviewer #2: No

3. Have the authors made all data underlying the findings in their manuscript fully available?

Reviewer #1: Yes

Reviewer #2: Yes

4. Is the manuscript presented in an intelligible fashion and written in standard English?

Reviewer #1: Yes

Reviewer #2: Yes

5. Review Comments to the Author

Reviewer #1: This is a well-written paper with a sound analysis, providing some interesting findings. The major contribution of this submission is that it is the first study in Ethiopia using viral load tests to confirm the correlates of non-adherence. Thus, while this study in Ethiopia identifies the barriers of adherence to first-line ART, there is not much new in the paper that has not been identified before. The method is sound and the study appears to have been nicely conducted. The results section is well-presented. I think this paper could be a publishable paper of some new insights within the Ethiopian context. However, I don't believe that PLOS one is the most appropriate venue as the journal includes pieces with broader focus and this paper may be a bit narrow. I would suggest that it should be published in a regional journal within Africa.

Reviewer #2: Ethical approval is not mentioned in the manuscript.

Detailed comments is as follows:

Comments to the paper: PONE-D-21-13362

General

The comments are described in the main manuscript with high-lighted texts and accessible to the author for possible amendments.

The paper is well articulated and relevant in addressing the current global health issue still imposing challenges to the globe even though showing promised changes.

Specifically to mention some,

Title

o Make it shorten if possible and to include hospital’s name

Introduction

o Abbreviated words/phrases should be explained fully at first appearance in the sentence, like ART, WHO, and the like.

o Distance of the study area from Addis Ababa, capital city of Ethiopia, not mentioned

o CD4 and viral load variables, how are they collected and analyzed, machine used never explained under data collection procedure

Methodology

o Study area is not fully described, distance/location from capital city not known.

o Sample size calculation not elaborated

o Ethical clearance and consent sub-section not included including approval board and number.

o Terms are not defined like adult, adherence (good, fair and poor), and the like

Result

o Table 1; others should be rechecked as it is greater than others; or else all mentioned variables should be listed out.

o Table2: p-value should be different from zero.

o Table 3: smoking and khat variables should be removed from that table. And also look for punctuation correction beneath the table

Discussion

o It becomes only one-sided discussion, only issue of similarity/consistency. Sometimes, disagreement is expected.

o Very recently published data/article (2021) not included in the study.

Conclusion

o It lacks recommendation

Limitation

o Not included as if the study is without limitation.

References

o Even though few, recent published data /online statistics for HIV-AIDS epidemiology not used/included.

6. PLOS authors have the option to publish the peer review history of their article (what does this mean?). If published, this will include your full peer review and any attached files.

Reviewer #1: No

Reviewer #2: No

---

## [Author Response · Author response to Decision Letter 0]

14 Sep 2021

Date 14/09/2021

To: PLOS ONE

Issue: Rebuttal Letter

Dear all at the PLOS ONE and esteemed reviewers, we are very grateful to your academic inputs. We found all your comments and editorial recommendations very important for betterment of our paper. In regard to this, we prepared a rebuttal letter responding to all your comments separately for journal requirement and for the two reviewers. The specific editorial and review comments are fully accepted and changes are made in the manuscript accordingly. The responses are placed next to the provided comments for all in the next pages of this letter. We are ready to further refine our manuscript if you found some of your comments are insufficiently accommodated.

Regards

Sabit Zenu Siraj

Corresponding Author of the Manuscript

 

Response to Journal Editorial Comments

Response: The document is revised to fulfil PLOS ONE’s style and file naming requirements.

Response: The required Ethics Statement is added to the manuscript file and to the online system. Manuscript, Page 9, Line 214-220.

3. We noted several instances in the manuscript where the P value equals zero; please correct and clarify.

Response: The P-Value in the table is corrected as P0.01. Manuscript, Page 11, Table 3

4. We note that you have indicated that data from this study are available upon request. PLOS only allows data to be available upon request if there are legal or ethical restrictions on sharing data publicly.

Response: The data availability statement is changed and the data is currently supplemented as supplementary file.

Response: The required Ethics Statement is added to the manuscript file and to the online system. Manuscript, Page 9, Line 214-220.

6. During our internal evaluation of the manuscript, we found significant text overlap between your submission and the following previously published works, some of which you are an author.

Response: The observed similarity with the said online sources is accidental. The observed similarity is currently paraphrased. Manuscript, Page 4, Line 84-88

 

Response to Reviewer 1

Reviewer Comments

This is a well-written paper with a sound analysis, providing some interesting findings. The major contribution of this submission is that it is the first study in Ethiopia using viral load tests to confirm the correlates of non-adherence. Thus, while this study in Ethiopia identifies the barriers of adherence to first-line ART, there is not much new in the paper that has not been identified before. The method is sound and the study appears to have been nicely conducted. The results section is well-presented. I think this paper could be a publishable paper of some new insights within the Ethiopian context. However, I don't believe that PLOS one is the most appropriate venue as the journal includes pieces with broader focus and this paper may be a bit narrow. I would suggest that it should be published in a regional journal within Africa.

Response: Dear Reviewer, we appreciate your encouragement and it will help us in our future endeavours. Regarding the suitability of PLOS ONE for this particular work, we submitted it to this journal because the journal is one of the most reputable. It publishes high-quality papers that have undergone extensive editorial and peer reviews. In Ethiopia, senior researchers actively encourage novice researchers like me to publish their works on PLOS ONE. In addition, the journal published high volume of academic researchers from Africa. According to the current data, the journal published over 5000 researches of different types conducted in or by Africans. Considering this, we would be happy if this paper is published on PLOS ONE.

 

Responses to Reviewer 2

Dear Reviewer, we are very grateful to your academic inputs. We amended the document as per your comments and specific responses are listed hereunder.

Title 

1. Make it shorten if possible and to include hospital’s name

Response: The title is shortened and the name of the hospital is included as ‘Determinants of first-line antiretroviral treatment failure among adult patients on treatment in Mettu Karl Specialized Hospital, South West Ethiopia; a case control study’. Manuscript, Title, Page: line 1-3

Introduction: 

2. Abbreviated words/phrases should be explained fully at first appearance in the

Response: All abbreviation issues are corrected. Manuscript, Introduction, Page 3-5, Line 52-126

3. Distance of the study area from Addis Ababa, capital city of Ethiopia, not mentioned

Response: The distance and some detail is added to the study area in Materials and Methods section. Manuscript, Materials and Methods, Page 6, Line 128-140.

4. CD4 and viral load variables, how are they collected and analysed, machine used never explained under data collection procedure

Response: The materials used in the hospital are added to the Materials and Methods sections under data collection tools and procedures. Manuscript; Materials and Methods, Data collection tools and procedures, Page 8, Line 189-195.

Methodology

5. Study area is not fully described, distance/location from capital city not known.

Response: The distance and other detail are added to the study area in Materials and Methods section. Manuscript; Materials and Methods, Page 6, Line 128-140.

6. Sample size calculation not elaborated

Response: The sample size was calculated by using Epi Info 7 using an option for Unmatched Case Control study. Power 80%, 95% CI and other parameters were used to calculate the sample size. Further detail is added to the document. Manuscript; Materials and Methods (sample size determination and sampling technique) Page 6-7, Line 154-178.

7. Ethical clearance and consent sub-section not included including approval board and number.

Response: The recommended Ethics Statement is added to the manuscript file and to the online system. Manuscript; Methods and Materials, Ethics Statement, Page 9, Line 214-220.

8. Terms are not defined like adult, adherence (good, fair and poor), and the like

Response: The terms are operationalized as recommended. Manuscript; Materials and Methods, Operational Definitions, Page 8, Line 197-203.

Result

9. Table 1; others should be rechecked as it is greater than others; or else all mentioned variables should be listed out.

Response: Corrected in the document. Manuscript; Result, Table 2, Page 9.

10. Table2: p-value should be different from zero

Response: The P-Value in the table is corrected as P0.01. Manuscript, Page 11, Table 3, the table is updated to table 3 as one more table is added to elaborate the sample size calculation procedures.

11. Table 3: smoking and khat variables should be removed from that table. And also look for punctuation correction beneath the table

Response: The variables are removed. The recommended changes are made to punctuations beneath the table. Manuscript; Result, Table 4, Page 13.

Discussion

12. It becomes only one-sided discussion, only issue of similarity/consistency. Sometimes, disagreement is expected.

Response: The recommended differences are added to the discussion. Manuscript; Discussion Page 26, line 358-369

13. Very recently published data/article (2021) not included in the study

Response: Recent UNAIDS 2021 report is added to the document and the information is updated accordingly.

Conclusion

14. It lacks recommendation

Response: Recommendation is added to the conclusion, Page 17, and Line 379-384

Limitation

15. Not included as if the study is without limitation.

Response: Limitation is added. Page 16, Line 370-372

References

16. Even though few, recent published data /online statistics for HIV-AIDS epidemiology not used/included.

Response: Updated epidemiological findings added to references. Page 18 and 19

---

## [Decision Letter · Decision Letter 1]

11 Oct 2021

Determinants of first-line antiretroviral treatment failure among adult patients on treatment in Mettu Karl Specialized Hospital, South West Ethiopia; a case control study

PONE-D-21-13362R1

Dear Dr. Siraj,

We’re pleased to inform you that your manuscript has been judged scientifically suitable for publication and will be formally accepted for publication once it meets all outstanding technical requirements.

Kind regards,

Frank T. Spradley

Academic Editor

PLOS ONE

Reviewers' comments:

Reviewer's Responses to Questions

**Comments to the Author**

1. If the authors have adequately addressed your comments raised in a previous round of review and you feel that this manuscript is now acceptable for publication, you may indicate that here to bypass the “Comments to the Author” section, enter your conflict of interest statement in the “Confidential to Editor” section, and submit your "Accept" recommendation.

Reviewer #1: All comments have been addressed

Reviewer #2: All comments have been addressed

2. Is the manuscript technically sound, and do the data support the conclusions?

Reviewer #1: Yes

Reviewer #2: Yes

3. Has the statistical analysis been performed appropriately and rigorously? 

Reviewer #1: Yes

Reviewer #2: Yes

4. Have the authors made all data underlying the findings in their manuscript fully available?

Reviewer #1: Yes

Reviewer #2: Yes

5. Is the manuscript presented in an intelligible fashion and written in standard English?

Reviewer #1: No

Reviewer #2: Yes

6. Review Comments to the Author

Reviewer #1: I can see that the reporting of the paper is improved and more clarity is provided. My original concern was with regards to the paper reaching the right audience and the fit to PLOS ONE, given that the findings are new and previously unstudied in the Ethiopian context, not the international context. Should the authors and editors of PLOS ONE both be happy about the fit and intended reach, I do believe that the research is well conducted and worthy of publishing.

Reviewer #2: First of all, I thank the journal editorial team to give me the opportunity of reviewing again. I have assessed the paper in that almost all comments were addressed by the author in-charge of the research paper.

7. PLOS authors have the option to publish the peer review history of their article (what does this mean?). If published, this will include your full peer review and any attached files.

Reviewer #1: No

Reviewer #2: No

---

## [Editor Report · Acceptance letter]

14 Oct 2021

PONE-D-21-13362R1 

Determinants of first-line antiretroviral treatment failure among adult patients on treatment in Mettu Karl Specialized Hospital, South West Ethiopia; a case control study 

Dear Dr. Zenu:

I'm pleased to inform you that your manuscript has been deemed suitable for publication in PLOS ONE. Congratulations! Your manuscript is now with our production department. 

Kind regards, 

on behalf of

Dr. Frank T. Spradley 

Academic Editor

PLOS ONE